# In Vitro Cell Behavior and Antibiotic Activity under Sustained Release of Doxycycline-Loaded Poly(lactic-co-glycolic acid) Microspheres

**DOI:** 10.3390/antibiotics11070945

**Published:** 2022-07-14

**Authors:** Flavia Pedrini, Virgínia S. Nazato, Moema A. Hausen, Daniel Komatsu, Stela S. Peña, Ana Lídia M. Almeida, Fernanda J. C. Pirola, Marina P. Françoso, Eliana A. R. Duek

**Affiliations:** 1Postgraduate Program in Biotechnology and Environmental Monitoring (PPGBMA), Federal University of São Carlos (UFSCar), Sorocaba 18052-780, Brazil; flaviampedrini@gmail.com (F.P.); vinazato@gmail.com (V.S.N.); eliduek@pucsp.br (E.A.R.D.); 2Department of Surgery, Faculty of Medical Sciences and Health, Pontifical Catholic University of São Paulo (PUC/SP), Sorocaba 18030-070, Brazil; dkomatsu@pucsp.br (D.K.); stela.pena@hotmail.com (S.S.P.); anamarcon.almeida@gmail.com (A.L.M.A.); fernandajcp@gmail.com (F.J.C.P.); marinapfrancoso@gmail.com (M.P.F.); 3Postgraduate Program in Materials Sciences (PPGCM), Federal University of São Carlos (UFSCar), Sorocaba 18052-780, Brazil

**Keywords:** drug-delivery, doxycycline, PLGA, antimicrobial activity, cell viability

## Abstract

The state-of-the-art sustained drug delivery systems are related to features to improve pharmacological transport through a controlled ratio between drug release and the desired therapeutic effect. Microspheres of biodegradable polymers, such as poly(lactic-co-glycolic acid) (PLGA), play an important role in these approaches, directing the release in a specific region of interest. In this way, the encapsulation of doxycycline (DOX) as a microbial agent turns the PLGA microspheres into a potential device for the treatment of topic oral diseases. Thus, this work aimed to produce DOX-loaded PLGA microspheres and see how they interfered with mesenchymal stem cell viability and in the sustained release in antimicrobial assays. Scanning electron microscopy showed the spherical microstructured pattern, revealing assorted sized distribution, with major diameters ranging 1–3 µm. The encapsulation efficiency presented a mean of 80% in both methods to obtain the microspheres (sonication and magnetic rotation). The DOX release test revealed a gradual and continuous profile of 30–40% between 120 and 168 h. Mesenchymal stem cells cultured in PLGA with or without DOX at several concentrations revealed no effect on the cell metabolic activity. Striking morphology changes were observed by confocal microscopy after 1 to 3 days under culture. The live/dead assay indicated that when microsphere densities were increased (from 10 to 100 µg/mL) cultured cells presented an internalized pattern of microspheres in both groups of PLGA containing DOX or not, while slight cell death signals were identified nearby microsphere clusters. Microbiological assays performed by the agar diffusion test pointed out that an inhibition zone was identified in *Staphylococcus aureus (S. aureus)* cultures at earlier times of DOX release. Despite the well-known use of PLGA as a drug delivery vehicle, when synthesized with DOX, it presents both characteristics of the desired treatment to prevent healthy tissue damage while avoiding bacterial growth in a microenvironment with anatomical features, such as grooves, projections, and other tough conditions that favor the development of oral diseases.

## 1. Introduction

Sustained release systems are designed to allow the introduction of therapeutic substances into the body and improve their safety and efficacy in a controlled and targeted pathway [1,2]. Microspheres of biodegradable polymers, such as poly(lactic-co-glycolic acid) (PLGA), play an important role in these approaches due to the material’s biocompatibility and degradation rate [3]. Its association with several drug types has been widely researched and the results demonstrate its potential as a drug carrier [4,5,6,7,8,9]. The key composition of microspheres lies in the application of interest; thus, to achieve the desired therapeutic effect, some features are decisive, including the monomer ratio, molecular weight, drug encapsulation, microsphere size, swelling properties, and morphology [10]. General literature content about PLGA usually applies the monomer ratio of 50:50 LA:GA with many physical-chemical characterizations that relate its potential as a drug carrier. However, this ratio is usually attributed to present a burst release profile and earlier degradation rates when compared to the ratios 90:10 or 80:20 [11]. In this context, in what concerns PLGA with higher LA ratio, there are scarce data that evaluate its application as a drug carrier.

Doxycycline (DOX) is a broad spectrum member of the tetracycline antibiotics, known for its fast bacterial action in Gram-negative and -positive protein synthesis. Oral administration of DOX is usually repeated several times per day to keep within the therapeutic range. Hence, local delivery of DOX using polymeric carriers is a reliable method of drug delivery for abroad applications, using assorted bioresorbable polymer-based carriers, such as chitosan, poly (vinyl alcohol), alginate, and gelatin [12,13].

One important issue on the topic of drug administration is the matter that it can generate high drug concentration levels in a microenvironment of inflamed tissues, due to the burst drug effect that commonly occurs in the first hours, which can induce local toxicity [14]. In this context, in vitro studies of controlled assessment to evaluate the direct cytotoxicity effect can enlighten how the sustained release may interfere with cellular activity as a whole. Thus, for the treatment improvement of oral infected diseases, DOX-loaded PLGA microspheres were produced by the single emulsion/solvent evaporation method and evaluated in vitro. As regards the cell behavior and antibiotic activity in *S. aureus*, this drug loaded in the polymeric system developed and achieved an early antimicrobial effect, while no toxicity level was identified in mesenchymal stem cells.

## 2. Materials and Methods

### 2.1. Preparation of PLGA and PLGA + DOX Microspheres

The microspheres were prepared using the single emulsion/solvent evaporation method, adapted from Terukina et al. (2016) [15]. For the PLGA + DOX microspheres, 10 mg of doxycycline (DOX) and 500 mg of poly(lactic acid-co-glycolic acid) (PLGA) 80:20 (Mw = 417,829 g/mol) were dissolved in 12 mL of dichloromethane on a magnetic stirrer. The mixture was dropped into 300 mL of 0.25% (*w*/*v*) poly(vinyl alcohol) (PVA) solution and then homogenized for 3 min at 10,000 rpm in an Ultra Turrex mixer (T25, IKA, Staufen, Germany). Before the solvent evaporation step, half of the obtained emulsion was sonicated for 10 min at 85% potency, setup at 180 W (UP 200 S, Hielscher). The other half was directly stirred to evaporate the solvent for 24 h at room temperature. Finally, both emulsions were centrifuged at 5000 rpm for 5 min and washed with distilled water (5804 R, Eppendorf, Hamburg, Germany). The microspheres (precipitate) were frozen at −20 °C for 24 h and lyophilized for an additional 24 h at −100 °C (K105, Liotop, São Paulo, Brazil). The samples obtained were submitted to a selective mesh of 45 µm to obtain more uniform size microsphere. The same protocol was performed for the PLGA microspheres without the addition of DOX. All reagents were purchased from Sigma-Aldrich (St. Louis, MO, USA).

### 2.2. Microspheres Characterization

#### 2.2.1. Fourier Transform Infrared (FT−IR)

FT-IR spectral data were obtained on a spectrometer (Spectrum 65, PerkinElmer Ltd., São Paulo, Brazil) with attenuated total reflection (ATR) to confirm Doxycycline encapsulation in the PLGA microspheres. The spectral scan was obtained in the range between 4000 and 500 cm^−1^.

#### 2.2.2. Morphology and Particle Size Distribution

The morphology of the microspheres was observed by scanning electron microscopy (SEM) (EVO MA15-ZEISS). Both PLGA and PLGA + DOX samples were submitted to a gold-sputter of 15–30 nm (Emitech K550) and visualized by secondary electrons at 15 kV. Randomly selected electron micrographs at 3000× magnification were used to obtain the microspheres diameter distribution by Image J software, counted up to reach the total number of 1000 counts.

#### 2.2.3. Encapsulation Efficiency (EE) and Drug Loading (DL)

The EE and DL were determined according to Equations (1) and (2), respectively [16]. The supernatant (1.5 mL) was analyzed by UV–VIS spectrophotometry (Cirrus 80, Femto Industries Ltd., São Paulo, Brazil) at 347 nm, from a standard curve containing a known content of DOX (Figure 1).
(1)$$EE\;\left(\%\right)=\frac{Amount\;of\;DOX\;measured\;in\;microspheres}{Amount\;of\;microspheres\;used\;for\;release}\times 100$$
(2)$$DL\;\left(\%\right)=\frac{Amount\;of\;DOX\;measured\;in\;microspheres}{Amount\;of\;DOX\;initially\;added\;to\;the\;formulation}\times 100$$

#### 2.2.4. Doxycycline Release Test

The in vitro release of DOX from the microspheres was determined as proposed by Misra et al. (2009) [17]. A total of 10 mg of DOX-containing microspheres were suspended in 3 mL of 0.01 M saline phosphate buffer (PBS) at 37 °C. The suspension was divided equally into two Eppendorf^®^ tubes (1.5 mL) under continuous shaking at 300 rpm (Thermomixer comfort, Eppendorf^®^, Hamburg, Germany). At specific time intervals (3, 6, 9, 24, 30, 48, 54, 72, 78, 96, 144, and 168 h), the tubes were centrifuged at 13,800 rpm at 4 °C for 10 min (Eppendorf^®^ 5804 R, Hamburg, Germany), and the supernatant was collected. The same amount of fresh PBS was added to each sample for the next readings in a UV–VIS spectrophotometer (Cirrus 80, Femto Industries Ltd., São Paulo, Brazil) at 347 nm. The cumulative release profile was calculated by the calibration curve obtained previously. No external stimuli (like temperature, pH or electrical changes) were conducted.

### 2.3. In Vitro Antimicrobial Activity

Doxycycline-loaded PLGA microspheres were tested for their antimicrobial properties on *S. aureus* by the agar diffusion test, according to the Reinbold et al. (2016) protocol [18]. The doxycycline-loaded microspheres were incubated in PBS, for 3, 6, 9, 24, 30, 48, 54, 72, 78, 96, and 100 h. After these specific periods, filter paper disks were soaked with 40 µL of the supernatant. As controls, commercial disks of gentamicin and doxycycline were used. The controls were not submitted to PBS incubation before seeding. The cultures of *S. aureus* were adjusted to 0.5 McFarland standards and seeded on Mueller–Hinton agar plates. Subsequently, the disks were placed on the plate and incubated for 24 h. The diameter of the inhibition zone was correlated to the bacterial sensitivity according to a susceptibility pattern obtained from reference [19].

### 2.4. In Vitro Biocompatibility

To exclude any influence on cell behavior when under contact with the microspheres, a wide range of concentrations of PLGA containing or not DOX was evaluated. In this way, to respond to whether the density could interfere with cell metabolism, the (3-(4,5-dimethylthiazol-2-yl)-2,5-diphenyltetrazolium bromide (MTT) assay was performed (Section 2.4.1), while to evaluate the survival rate and any morphological modification, the cells were observed using a live/dead kit (Section 2.4.2). The PLGA or PLGA + DOX powder was titrated in supplemented Dulbecco’s modified Eagle medium (DMEM) at the concentrations of 0.1, 1, 5, 50, and 100 µg/mL. The human Mesenchymal Stem Cells (hMSC) were purchased from Thermo Fisher Scientific (Stem Pro^TM^ Human Adipose-Derived Stem Cell Kit) and cultured according to the kit protocol.

#### 2.4.1. Viability by MTT Assay

After 1 day in culture in 96-well plates, the hMSC cells medium was exchanged for the extract medium in all concentrations of PLGA and PLGA + DOX. Then, the subconfluent culture was cultivated for an additional period of 24 h and 72 h. To evaluate cell metabolism, the medium was exchanged to DMEM containing MTT at 0.05% for 2 h, washed with saline solution, and lysed with dimethylsulfoxide. The formed formazan content was revealed and read at 570 nm at the Biotex Ex800 microplate reader.

#### 2.4.2. Cell Survival and Morphology

The assay was performed when cells reached passages between three and five. Before each assay, both PLGA and PLGA + DOX microspheres were UV-C sterilized. The hMSC were seeded at 2 × 10^4^ cells/well. All samples submitted to microscopy were cultured in Thermanox^®^ coverslips (Nunc Thermo Scientific, Rochester, NY, USA). Cell proliferation and viability were evaluated after 1 and 3 days, as described by Komatsu et al. (2019) [20]. Cells were subjected to fluorescence live imaging (Live/Dead KIT) and analyzed by laser scanning confocal microscopy (LSCM) (model TCS SP8; Leica Microsystems GmbH, Heidelberg, Germany). The system was set up to detect signals using photomultiplier tube mode (PMT) mode with lasers line 488 and 647 nm to identify Calcein AM and EthD-1, respectively. The reflection images were accessed by the PMT Trans detection mode.

### 2.5. Statistical Analysis

The particle size distribution was analyzed by Gaussian curves and indicates mean size. The data of the MTT assay were submitted to the one-way ANOVA parametric Tukey statistical test. The results are represented by the mean ± standard deviation.

## 3. Results

During microsphere preparation, the emulsion was obtained by two different mixing mechanisms: sonication (S) and magnetic rotation (MR). Both methods tested revealed minimal and insignificant yield differences, as shown in Table 1.

### 3.1. FT−IR

The FT-IR spectra of PLGA and PLGA + DOX are depicted in Figure 2. The main functional groups of PLGA are identified by the peaks in the region of 1750 cm^−1^, which are assigned to the stretching vibration of the carbonyl groups (C=O) present in both monomers, and the 1089 cm^−1^ peak attributed to C–O stretching [21]. When PLGA + DOX spectra were analyzed, the band at 1615 cm^−1^, specific for the –COOH group present in doxycycline was identified [22]. The results showed that no chemical interaction between DOX and PLGA was established since the typical bands were maintained after the drug encapsulation.

### 3.2. Microsphere Measurements

All microspheres synthesized in this work were reproduced independently in sextuplicates, and the samples that reached higher encapsulation efficiency were selected for SEM. Both emulsion mixing methods presented as sonication (Sample 1) and magnetic rotation (Sample 3) are depicted in Figure 3. The morphological pattern revealed a perfect round shape with different sizes.

The size measurement revealed that the average microsphere size ranges from 1 to 3 µm (±50%), while the size range from 1 to 10 µm represents 90% of all particle diameters, as seen in Figure 4.

This result also demonstrated that different mixing mechanisms did not show a variation in the size of the microspheres produced.

### 3.3. Encapsulation Efficiency

Samples submitted to MR presented an encapsulation efficacy mean of 82.50 ± 2.85% while the S-prepared samples were 83.07 ± 1.54% (Table 2). The results demonstrated discrete loss of the drug (around 15%) during the process of obtaining PLGA microspheres.

### 3.4. Doxycycline Release Profile

A proportional release profile between samples was identified and the drug delivery increased with time (Figure 5 and Figure 6). The major release profile from samples was observed up to 72 h, and soon after, the curves tended to present a lower release rate. The endpoint showed that between samples, the release ranged around 20% in both mixing methods tested (MR and S).

### 3.5. In Vitro Antimicrobial Activity

The microbiological assays revealed a good distribution pattern of the inhibition zone up to 48 h under *S. aureus* culture (Figure 7). Longer cultures up to 100 h were also performed but no inhibition zone was identified (data not shown). The data was measured, and quantification is depicted in Table 3.

### 3.6. Cell Viability by Metabolic Activity

The hMSC were cultured for 1 and 3 days in DMEM supplemented with 10% Fetal Bovine Serum (FBS) containing PLGA and PLGA + DOX at several concentrations. All concentrations between both groups presented no statistical significance, while an increased absorbance signal was observed by time course due to cell growth (Figure 8).

### 3.7. Cell Survival and Morphology by Live/Dead Assay

The live/dead identification kit revealed that all groups presented high cell survival rates after 3 days in culture (Figure 9 and Figure 10). Few dead cells were observed. The microspheres were observed in reflection mode as dark-shadowed dots or as aggregates, deposited in coverslip surfaces or associated with the cells. When images were overlaid, it revealed that no fluorescent labeling was identified in the areas containing microsphere clusters, inducing changes in cell morphology not related to cell death.

## 4. Discussion

Several methods have been proposed to obtain PLGA microspheres. In the simple emulsion/solvent evaporation method, the emulsion can be prepared by assorted physical methods, including sonication and magnetic rotation [23]. It is known that the duration of sonication and stirring rate can influence the yield and size distribution of the microspheres [24]. Nonetheless, no significant differences between groups were identified when yield and size distribution were compared. Despite the different sizes observed by SEM (Figure 3), the microspheres presented as smooth spherical particles between 1–3 µm (Figure 4).

The FT-IR spectra identified the DOX presence (Figure 2) that can be correlated to the evaluation of encapsulation efficiency (Table 2), and both emulsion preparation methods did not show any difference. The overall gain of DOX encapsulation reached around 85% of efficiency. Mundargi et al. [25] pointed out that the simple emulsion method has the drawback of yielding low encapsulation efficiency. Cheng et al. (2006) [26] reached an encapsulation efficacy of 68% to 79% in PLGA loaded with magnetic protein of 4–7µm in a double emulsion method. The simple emulsion method used in this work presented similar efficacy in the same range of particle sizes. When a drug delivery system is used, it is known that the particle size influences directly in the release rate due to the surface area, which is related to the water-based carrier system to absorb the desired drug [27]. In this way, the release profile identified in Figure 5 and Figure 6 showed no drug burst in 24 h, but a progressive release pattern. Lin et al. (2018) [28] showed that small PLGA 50/50 microspheres of 3.8 µm presented a huge burst profile in vitro after 24 h. The equal ratio of L and G in PLGA is the most used monomer charge profile in the majority of studies that use PLGA as a drug carrier [29]. The increased ratio of lactic acid in PLGA changes the microsphere degradation to a lower rate [30,31]. According to Dawes et al. 2009 [32], the small surface area of PLGA leads to a reduced rate of water permeation and matrix degradation, and it is known that the particle size is determined by the method used and the monomer ratio. Thus, the PLGA 80/20 used in this work might have affected the early burst already described in the literature.

Despite the cumulative release rate identified in Figure 5 and Figure 6 for up to 7 days, this rate did not correlate to the DOX drug activity due to the huge loss of the in vitro microbiological activity after 48 h, as observed in Figure 7. Misra et al. (2009) [17] identified a loss in antimicrobial activity of DOX in *E. coli* cultures after 24 h and indicated that a sustained release formulation is required to control bacterial growth for longer periods. Zhong et al. (2022) [33] showed that DOX differs from other tetracycline drugs due to its higher hydrolytic degradation under alkaline pH and temperature range, with a half-life of 6.5 h under neutral pH at 25 °C. In this context, to maintain the bacteriological effect, the strategic outcome of effective treatment should be a repeated administration of PLGA/DOX every 24 h, independently of the continuous release rate over a long time.

The evaluation of PLGA or PLGA + DOX in hMSC cell culture was identified by metabolic activity measurement during MTT assay (Figure 8) and corroborated by live/dead LSCM observations (Figure 9 and Figure 10). When cell behavior was evaluated under the increased addition of the microspheres in the cell medium, no effect was identified up to the tested concentration of 100 µg mL^−1^ in PLGA and PLGA + DOX. However, despite there being no evidence of changes in cell growth, the LSCM showed differences in cell morphology. Interestingly, when aggregated microspheres were identified, the cellular organization also followed a pattern with a high amount of cells surrounding such clusters. This pattern was already reported as an early sign of spheroid formation elsewhere in gelatin microspheres [34], while the morphological cell modification surrounding microspheres of chitosan and fumarate-vinyl acetate was reported by Lastra et al. (2020) [35].

A similar work recently presented a sustained release pattern of DOX from PLGA microspheres obtained by a double-emulsion method, indicating that microspheres were not toxic, kept the sustained release for up to 21 days and were indicated as a potential periodontitis treatment [36]. In this way, due to the short half-life of DOX identified in this work and also reported in the literature, one could use this DOX-loaded microsphere model to treat periodontitis only during the DOX active period (24 h).

It is important to emphasize here that the use of PLGA + DOX, or even DOX with any other polymeric model of sustained delivery, has constraints due to the limited DOX activity time identified in microbiological assays. Although the release rate has prolonged features, for some treatments such as periodontitis, or even other oral diseases that form bacterial pockets or abscesses, a new DOX administration should be repeated every 24 h. Thus, in future in vivo works, it is necessary to evaluate the issue of a cumulative effect of the physical presence of the carrier biomaterial to be successfully recurrently added at the infection site until the infectious process is solved.

## 5. Conclusions

Both emulsion preparation methods used generated the same particle size of PLGA + DOX, reaching around 85% of encapsulation efficacy. The FT-IR revealed a physical association of DOX and PLGA, while no burst was identified in release tests. Assorted particle sizes identified by SEM revealed average sizes of 1–3 µm (50%) and 3–5 µm (10%). Microbiological tests were moderately effective up to 24 h to inhibit *S. aureus* growth. The cell viability assays demonstrated that after several concentrations tested of PLGA and PLGA + DOX, the metabolic activity remained similar to controls, while the live/dead survival assay showed differences in cell morphology due to adhesion surrounding microspheres aggregates, with no indication of cell death.

## Figures and Tables

**Figure 1 antibiotics-11-00945-f001:**
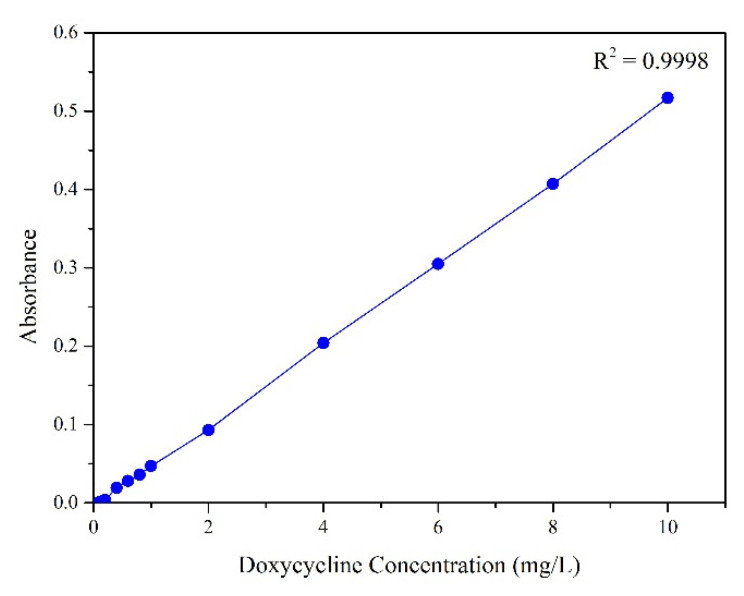
Standard curve obtained from known concentrations of DOX analyzed by UV–VIS spectrophotometry.

**Figure 2 antibiotics-11-00945-f002:**
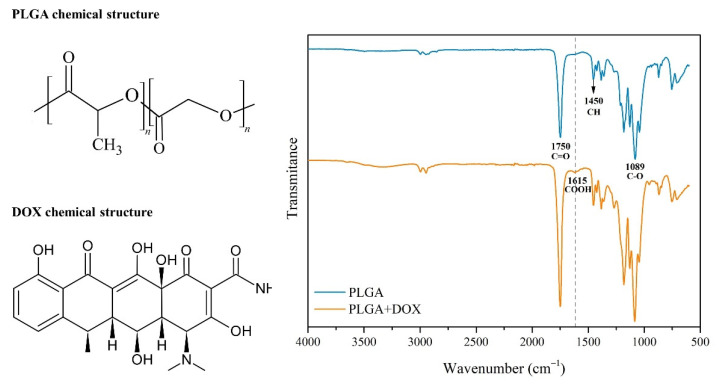
FT−IR spectra of PLGA and PLGA + DOX microspheres.

**Figure 3 antibiotics-11-00945-f003:**
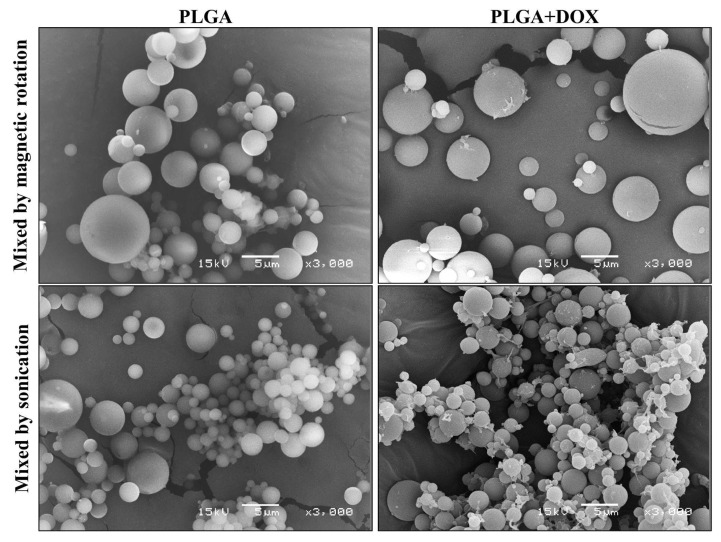
PLGA and PLGA + DOX microspheres observed by scanning electron microscopy. Bars 5 µm.

**Figure 4 antibiotics-11-00945-f004:**
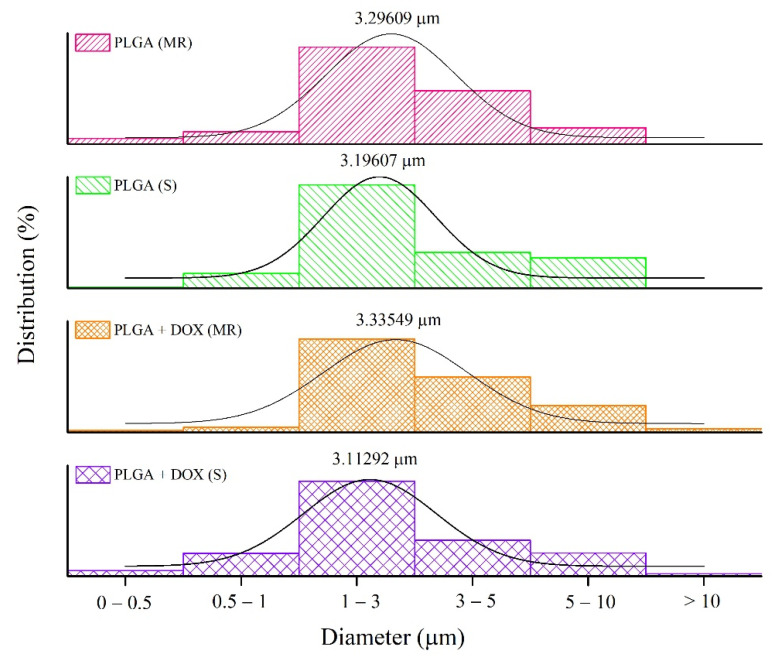
Particle size distribution of PLGA and PLGA + DOX microspheres submitted to sonication (S) and magnetic rotation (MR) (histogram and Gaussian fit).

**Figure 5 antibiotics-11-00945-f005:**
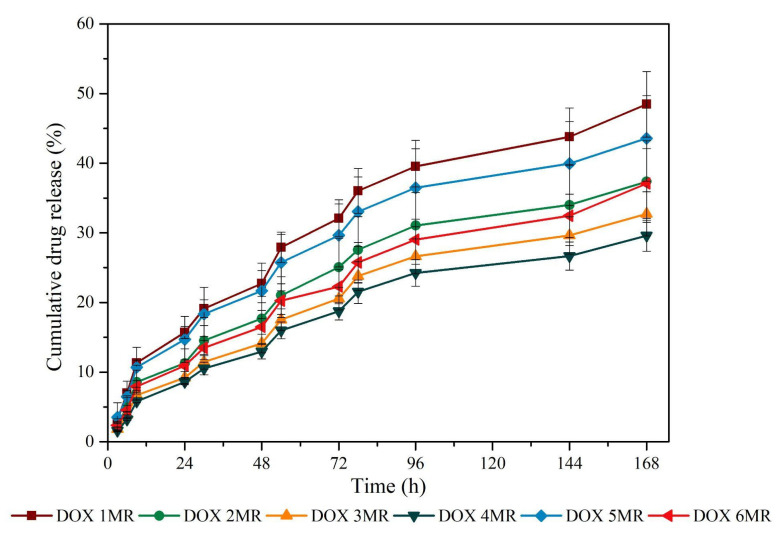
Release profile of all PLGA + DOX samples submitted to magnetic rotation (MR).

**Figure 6 antibiotics-11-00945-f006:**
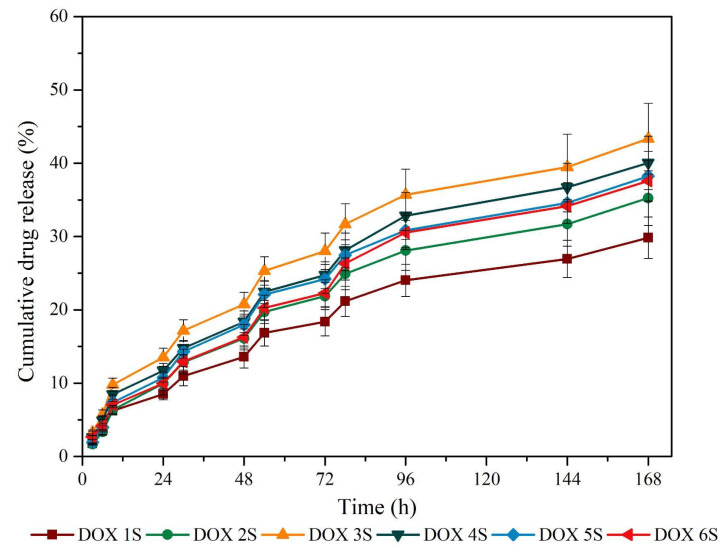
Release profile of all PLGA + DOX samples submitted to sonication (S).

**Figure 7 antibiotics-11-00945-f007:**
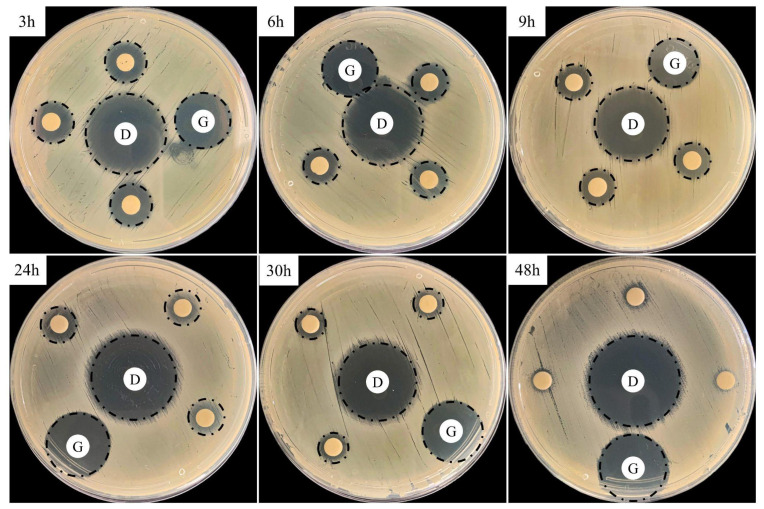
Microbiological assays obtained from the Agar-diffusion test after 24 h in culture. The triplicate of unnamed disks represents treated samples after microsphere drug release. D and G represent doxycycline and gentamicin commercial controls, respectively.

**Figure 8 antibiotics-11-00945-f008:**
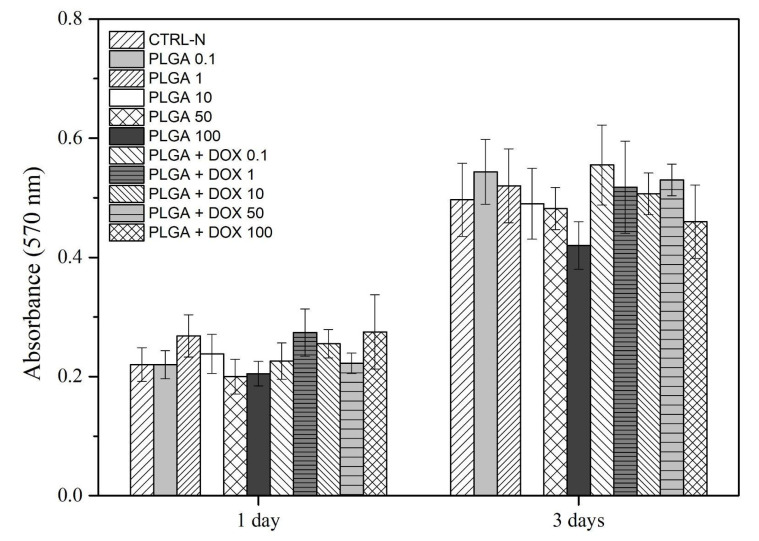
Absorbance detected by the MTT reduction colorimetric assay; mean values and standard deviation of mesenchymal stem cells cultured for 1 and 3 days in DMEM medium containing several concentrations of PLGA and PLGA + DOX.

**Figure 9 antibiotics-11-00945-f009:**
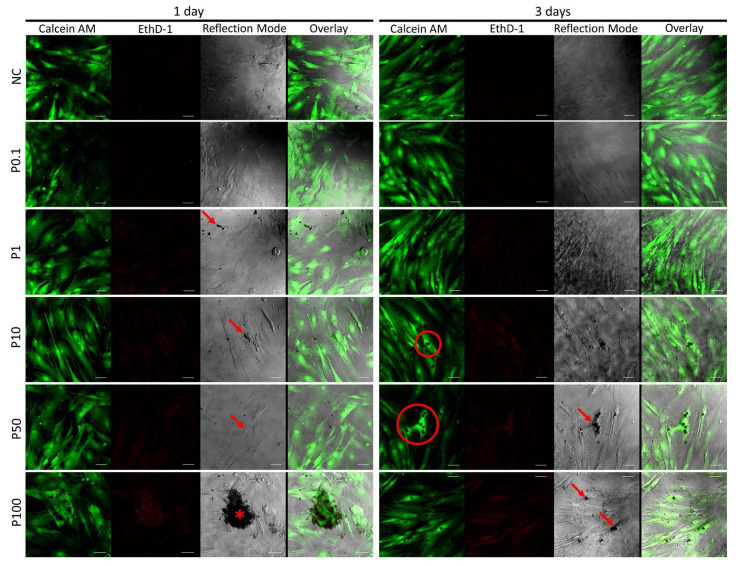
Confocal microscopy images of hMSCs cultured for 1 and 3 days in DMEM containing PLGA microspheres at concentrations of 0.1, 1, 10, 50, and 100 μg/mL identified as P0.1, P1, P10, P50, P100, respectively; the observation depicts live/dead cells (green and red, respectively), and small/large microsphere aggregates (arrows and asterisks, respectively). NC: negative control; bars 50 μm.

**Figure 10 antibiotics-11-00945-f010:**
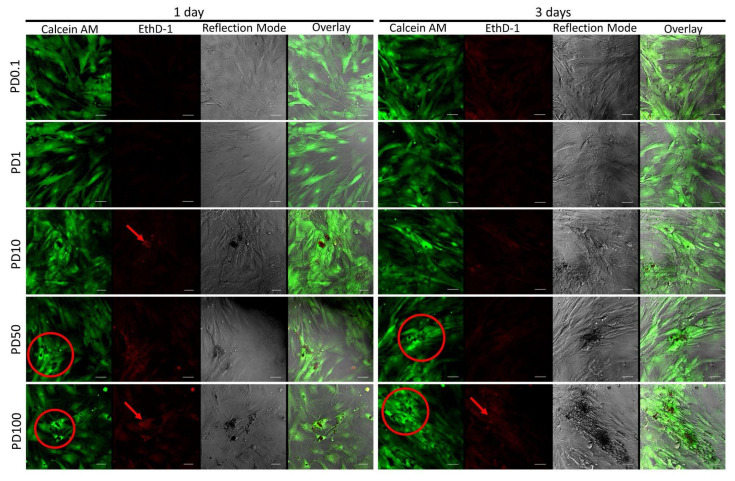
Confocal microscopy images of hMSCs cultured for 1 and 3 days in PLGA containing DOX at concentrations of 0.1, 1, 10, 50, and 100 μg/mL identified as PD0.1, PD1, PD10, PD50, and PD100, respectively; the observation depicts live/dead cells (green and red, respectively) and small and large microsphere aggregates (arrows and asterisks, respectively); bars 50 μm.

**Table 1 antibiotics-11-00945-t001:** Average yield of PLGA and PLGA + DOX microspheres mixed by sonication (S) and magnetic rotation (MR).

Sample	Yield
PLGA (S)	75.69 ± 3.71%
PLGA (MR)	77.04 ± 2.87%
PLGA + DOX (S)	71.69 ± 8.21%
PLGA + DOX (MR)	74.01 ± 4.87%

**Table 2 antibiotics-11-00945-t002:** Encapsulation efficiency and drug loading in PLGA + DOX microspheres.

Sample	Initial DOX Concentration (mg/L)	Unencapsulated DOX Concentration (mg/L)	Initial DOX Concentration *Minus* Unencapsulated DOX Concentration (mg/L)	Encapsulation Efficiency (%)	Drug Loading (%)
(MR)					
DOX 1	38.46	8.27	30.19	78.50	41.66
DOX 2	48.08	8.98	39.10	81.32	33.32
* DOX 3	51.28	6.69	44.59	86.96	31.25
DOX 4	54.49	8.71	45.78	84.01	29.41
DOX 5	41.67	7.71	33.96	81.50	38.47
DOX 6	44.87	7.77	37.10	82.69	35.70
(S)					
* DOX 1	54.49	7.73	46.76	85.82	29.41
DOX 2	44.87	7.52	37.35	83.25	35.71
DOX 3	38.46	7.11	31.35	81.51	41.62
DOX 4	41.67	6.94	34.73	83.35	38.44
DOX 5	41.67	7.57	34.09	81.82	38.44
DOX 6	44.87	7.79	37.09	82.65	35.70

* Crosshatched samples were selected for SEM analysis, while only DOX 3 MR was used to perform the biocompatibility assays.

**Table 3 antibiotics-11-00945-t003:** Evaluation of the inhibition zone diameter obtained in microbiological assays.

	Inhibition Zone (mm)			
	Controls		Treated Samples *	
	Doxycycline (D)	Gentamicin (G)	PLGA + DOX **	
Time (h)				Susceptibility ***
3	30	20	14	Intermediate
6			13	Intermediate
9				
24				
30			10	Resistant
48			7	Resistant

* Triplicate mean; ** Disk diameter was 6mm; *** According to the M100, 27th edition for use with M02-A12 and M07-A10 [19].

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
