# Peer review of "In Vitro Cell Behavior and Antibiotic Activity under Sustained Release of Doxycycline-Loaded Poly(lactic-co-glycolic acid) Microspheres"

_antibiotics, 2022, doi:10.3390/antibiotics11070945_

Round 1
Reviewer 1 Report
Dear colleagues,
Congratulation for your research on DDS with biodegradable polymers. the paper shows a lot of work.
I insert herein some infos that may help you improving the manuscript.
Section 1 - I suggest you add some arguments on analytical methods applied.
Section 2
2.1. - please add details related to sonication method used, frequency, instrument, etc. and, equipment used for lyophilization, centrifugations etc
- please add the system used for the infrared spectroscopy measurements (KBr pellets / ATR, transmission / reflection etc)
- please carefully check defining the acronyms at their first use in the text (i.e. hEMS, MTT, DMEM, PMT in 2.4. etc)
- for the UV-VIS spectrometric methods applied, details are needed, or references where these may be found (i.e. range of the calibration curve used, if dilutions of the samples were performed, etc) i.e. in 2.2.3., 2.2.4., 2.4.1.
Section 3
- fig. 2 - the scale not clearly readable for the 2 images corresponding to sonicated samples
- table 2 - the 4th coloumn seems showing redundant information; I suggest removal, and instead providing the calculation formula for the encapsulation efficiency
- FBS acronym definition in 3.5.
Reviewer 2 Report
In this study, the authors developed PLGA-based microspheres as the drug delivery system and evaluated it for the release of doxycycline (DOX). The drug loading in microspheres was confirmed through FTIR spectroscopy, and developed microspheres were characterized for various properties such as size, morphology, drug encapsulation efficiency, and drug release profile. Furthermore, the microspheres were tested for in vitro biocompatibility and antimicrobial activity. The study reports some interesting findings and is worthy of publication; however, there are a few issues that need to be addressed.
1. Authors determined the drug encapsulation efficiency (EE) but did not determine the drug loading (DL) efficiency of the developed microspheres. The drug loading efficiency can be determined with the below provided formula (Engineered Science. 2021, 13: 106-120). Also include the below formula for determining the drug encapsulation efficiency in the main manuscript.
EE (%) = Amount of drug measured in MS/Amount of drug initially added to the formulation×100
DL (%) = Amount of drug measured in MS/Amount of MS used for extraction ×100
2. Many studies have reported the development of PLGA microspheres for drug delivery applications. Please clarify the novelty of the study in the Introduction section.
3. Can the developed PLGA-based microspheres drug delivery system be applied for the release of other drugs?
4. Introduction section is too weak. Here authors mainly talked about PLGA microspheres. IT is suggested to add a discussion on the development of other polymers-based microspheres used for drug delivery for comparison (Biomaterials, 2022, 285: 121530; Materials Science and Engineering C. 2021, 121:111889; Materials Science and Engineering C. 2020, 115: 111107; Bioactive Materials. 2021, 6:2105–2119).
5. Does the drug release require any external stimuli, like temperature, pH, or electrical stimulation?
6. Did you determine the swelling behavior of the microspheres? This is an important feature of the drug release behavior of the microspheres.
7. Figure 6. A negative control (PLGA without DOX) is missing.
8. Section 2.1: Did you optimize the ratio of PLGA and DOX for the preparation of composite microspheres?
9. Figure 2. SEM micrographs show a clear size variation of the developed microspheres. Can we control the size? I suggest using standard sieves (of selected mesh) to obtain more uniform size microspheres for different analyses (Biomaterials Science. 2020, 8: 2797-2813).
10. I suggest combining the ‘Results’ and ‘Discussion’ sections as it will allow the readers to read the results and associated discussion without moving back and forth.
11. Number the Section ‘FTIR’ under the ‘Results’ section (below Table 1).
12. Full form of microbial species (S. aureus) should be provided at first appearance.
13. Change minutes to min, hours to h, and so on.
14. Few grammatical errors and typos exist in the manuscript, and thus careful proofreading is required at the revision stage.
Round 2
Reviewer 2 Report
Accept